# Convalescent COVID-19 patients are susceptible to endothelial dysfunction due to persistent immune activation

**Florence WJ Chioh[1†], Siew-Wai Fong[2,3,4†], Barnaby E Young[1,5,6], Kan-Xing Wu[1], Anthony Siau[1], Shuba Krishnan[1,7], Yi-Hao Chan[2,3], Guillaume Carissimo[2,3], Louis LY Teo[8,9], Fei Gao[8,9], Ru San Tan[8,9], Liang Zhong[8,9], Angela S Koh[8,9], Seow-Yen Tan[10], Paul A Tambyah[11], Laurent Renia[1,2,3], Lisa FP Ng[2,3], David C Lye[1,5,6,12], Christine Cheung[1,13]***

[1]Lee Kong Chian School of Medicine, Nanyang Technological University, Singapore, Singapore; [2]A*STAR ID Labs, Agency for Science, Technology and Research, Singapore, Singapore; [3]Singapore Immunology Network, Agency for Science, Technology and Research, Singapore, Singapore; [4]Department of Biological Sciences, National University of Singapore, Singapore, Singapore; [5]National Centre for Infectious Diseases, Singapore, Singapore; [6]Department of Infectious Diseases, Tan Tock Seng Hospital, Singapore, Singapore; [7]Division of Clinical Microbiology, Department of Laboratory Medicine, Karolinska Institute, ANA Futura, Campus Flemingsberg, Stockholm, Sweden; [8]National Heart Centre Singapore, Singapore, Singapore; [9]Duke-NUS Medical School, Singapore, Singapore; [10]Department of Infectious Diseases, Changi General Hospital, Singapore, Singapore; [11]Department of Medicine, National University Hospital, Singapore, Singapore; [12]Yong Loo Lin School of Medicine, National University of Singapore, Singapore, Singapore; [13]Institute of Molecular and Cell Biology, Agency for Science, Technology and Research, Singapore, Singapore

**\*For correspondence:**
ccheung@ntu.edu.sg

[†]These authors contributed equally to this work

**Abstract** Numerous reports of vascular events after an initial recovery from COVID-19 form our impetus to investigate the impact of COVID-19 on vascular health of recovered patients. We found elevated levels of circulating endothelial cells (CECs), a biomarker of vascular injury, in COVID-19 convalescents compared to healthy controls. In particular, those with pre-existing conditions (e.g., hypertension, diabetes) had more pronounced endothelial activation hallmarks than non-COVID-19 patients with matched cardiovascular risk. Several proinflammatory and activated T lymphocyte-associated cytokines sustained from acute infection to recovery phase, which correlated positively with CEC measures, implicating cytokine-driven endothelial dysfunction. Notably, we found higher frequency of effector T cells in our COVID-19 convalescents compared to healthy controls. The activation markers detected on CECs mapped to counter receptors found primarily on cytotoxic CD8+ T cells, raising the possibility of cytotoxic effector cells targeting activated endothelial cells. Clinical trials in preventive therapy for post-COVID-19 vascular complications may be needed.

## Introduction

As of February 21, 2021, there have been more than 110 million confirmed cases and 2.4 million deaths from coronavirus disease 2019 (COVID-19). Many countries are facing multiple waves of resurgence upon reopening their economy, prompting new lockdown. Intensive ongoing research has shed light on the pathogenesis of COVID-19 and the extent of damages caused directly or

indirectly by severe acute respiratory syndrome coronavirus 2 (SARS-CoV-2). Conversely, the intermediate and long-term complications of COVID-19 remains unclear (*del Rio et al., 2020*). While most infected people recover completely within a few weeks, a considerable proportion continue to experience symptoms after their initial recovery (*Yelin et al., 2020*), similar to SARS survivors (*Ngai et al., 2010*).

COVID-19 is mainly a respiratory infection. However, both autopsy findings and clinical observations have described vascular damages and thrombotic complications in a wide range of organs (*Gupta et al., 2020*; *Price et al., 2020*; *Xu et al., 2020*). Endothelialitis may occur in multiple organs as a consequence of viral infection and overactive host immune response as T cell infiltration was observed in sites of endothelial damage of COVID-19 patients' tissues. Systematic evaluation of the long-term sequelae of COVID-19 is currently lacking. COVID-19 infection can cause extrapulmonary pathologies that persist after recovery such as cardiovascular events with ongoing myocardial inflammation (*Puntmann et al., 2020*), acute ischemic or hemorrhagic stroke (*Mao et al., 2020*), as well as liver injury, neurological deficits, and acute kidney injury necessitating dialysis (*Gupta et al., 2020*). The thrombotic complications of COVID-19, such as pulmonary embolism may lead to lasting organ failure (*Price et al., 2020*). Independent of acute respiratory distress syndrome, severe pneumonia has been consistently associated with augmented risk of cardiac impairment both during convalescence and in later years (*Corrales-Medina et al., 2015*). We hypothesize that the risk of vascular complications in COVID-19 survivors will be higher, if compounded by a confluence of SARS-CoV-2-mediated damages, proinflammatory cytokine overdrive, and comorbidities with hypertension and diabetes.

Circulating endothelial cells (CECs), dislodged from blood vessels as a consequence of vascular injury, constitute an ideal cell-based biomarker as it is indicative of in situ endothelial pathophysiology (*Farinacci et al., 2019*; *Hill et al., 2003*). Although CECs are a rare population in peripheral blood, they reflect endothelial dysfunction from a variety of vascular disorders, including myocardial infarction, acute ischemic stroke, atherosclerosis, and vasculitis (*Nadar et al., 2005*; *Schmidt et al., 2015*). In COVID-19, there are conflicting reports as to the frequency of CEC counts compared with healthy controls (*Mancuso et al., 2020*; *Nizzoli et al., 2020*). This discrepancy may reflect disease severity as COVID-19 patients admitted to the intensive care unit (ICU) have a higher CEC count, suggesting more pronounced endothelial injury in severe COVID-19 (*Guervilly et al., 2020*). Interestingly, among non-ICU patients, those with chronic kidney disease had significantly higher CEC counts, implying that patients with pre-existing conditions may be more susceptible to vascular damage (*Guervilly et al., 2020*).

This study aims to understand the state of vascular health in convalescent COVID-19 patients and to evaluate subclinical endothelial dysfunction through the phenotyping of CECs.

## Results

### Patient and healthy participants characterization

To understand the intermediate consequence of COVID-19, we performed vascular phenotyping using CECs and endothelial activation markers as the cellular and molecular measures of endothelial dysfunction. Written informed consent was received from participants prior to inclusion in the PROTECT study (*Young et al., 2020a,c*). All study groups were almost gender-balanced, with prior comorbidities and self-identified ethnicity/race as summarized in *Table 1*. Convalescent COVID-19 patients who had no pneumonia throughout admission (mild), pneumonia without hypoxia (moderate), or pneumonia with hypoxia (desaturation to ≤94%) requiring supplemental oxygen (severe), and discharged after 15 days (median [interquartile range, IQR 11–26]) were screened for pre-existing cardiovascular risk factors. Among our selected convalescent COVID-19 patients (n = 30), half had at least one cardiovascular risk factor, including mainly hypertension, diabetes, and/or hyperlipidemia. They were benchmarked to healthy participants (n = 24) and non-COVID patients with matched cardiovascular risk factors (n = 20). More details on demographics are included in *Table 1*.

**Table 1.** Demographics of patients and healthy controls.

| Characteristics, N (%) | Healthy participants (n = 24) | Non-COVID-19 patients with cardiovascular risk factors (n = 20) | Convalescent COVID-19 individuals (n = 30) | |
| --- | --- | --- | --- | --- |
| | | | Cardiovascular risk factors (n = 15) | No cardiovascular risk (n = 15) |
| Age* | 46.5 (9.9) years | 62.4 (8.7) years | 54 (8.7) years | 42 (13.5) years |
| Age, male* | 46.3 (11.3) years | 62.3 (10.4) years | 54 (8.2) years | 36 (6.1) years |
| Age, female* | 46.8 (8.8) years | 63 (7.3) years | 54 (9.9) years | 48 (17.1) years |
| Gender, male | 12 (50) | 10 (50) | 8 (53.3) | 8 (53.3) |
| Gender, female | 12 (50) | 10 (50) | 7 (46.7) | 7 (46.7) |
| Ethnicity | Chinese 22 (91.7) | Chinese 17 (85) | Chinese 11 (73.3) | Chinese 13 (86.6) |
| | Filipino 2 (8.3) | Indian 3 (15) | Malay 2 (13.3) | Indian 1 (6.7) |
| | | | Bangladeshi 1 (6.7) | Caucasian 1 (6.7) |
| | | | Filipino 1 (6.7) | |
| Days post-symptom onset (median [IQR]) | N.A. | N.A. | 34 (27–46) days | |
| COVID-19 severity | N.A. | N.A. | Mild 5 (33.3) | Mild 5 (33.3) |
| | | | Moderate 5 (33.3) | Moderate 5 (33.3) |
| | | | Severe 5 (33.3) | Severe 5 (33.3) |
| Comorbidities | | | | |
| Hypertension | 0 (0) | 20 (100) | 10 (66.7) | 0 (0) |
| Hyperlipidemia | 0 (0) | 10 (50) | 11 (73.3) | 0 (0) |
| Diabetes mellitus | 0 (0) | 9 (45) | 7 (46.7) | 0 (0) |
| Fatty liver | 0 (0) | Information not available | 2 (13.3) | 0 (0) |
| Chronic liver disease | 0 (0) | Information not available | 1 (6.7) | 0 (0) |
| Coronary artery disease | 0 (0) | 6 (30) | Information not available | Information not available |
| Myocardial infarction | 0 (0) | 0 (0) | 2 (13.3) | 0 (0) |

All values are reported as N (%) where N indicates the number of observations.

*Values are expressed as mean (± standard deviation).

## COVID-19 and cardiovascular risk factors contribute to endothelial dysfunction

There are two major types of blood endothelial cells (*Hebbel, 2017*). CECs are shed from damaged vessels and constitute a cell-based biomarker for vascular dysfunction (*Blann et al., 2005*). On the other hand, endothelial progenitor cells (EPCs) originate from the bone marrow and are mobilized into the bloodstream in response to vascular injury (*Ackermann et al., 2020a,bAckermann et al.,c; Li and Li, 2016; Mancuso et al., 2020*). Identification of the CEC population from individual peripheral blood mononuclear cells (PBMCs) samples was carried out with high stringency using a panel of established CEC immunophenotypic markers (*Figure 1a*). We first gated for the CD45$^-$/CD31$^+$ population to isolate CECs along with bone marrow-derived EPCs and platelets, while ruling out CD45$^+$ leukocytes that also expressed the endothelial marker CD31 (PECAM1) (*Duda et al., 2007*). This population was further subtyped based on the progenitor marker, CD133, to rule out EPCs. Finally, a nucleic acid stain was used to distinguish anucleate platelets from nucleated CECs, which were defined here by a combined immunophenotypic profile of CD45$^-$/CD31$^+$/CD133$^-$/DNA$^+$ (*Burger and Touyz, 2012; Duda et al., 2007; Yu et al., 2013*). As a comparison, putative EPCs were identified by CD133$^+$/CD45$^-$/DNA$^+$ population (*Ingram et al., 2004; Mancuso et al., 2020; Rafii and Lyden, 2003*). To further analyze endothelial cell activation, CECs were characterized for the expression of activation markers, namely, intercellular adhesion molecule 1 (ICAM1), P-selectin (SELP), and fractalkine (CX3CL1) (*Figure 1a*), which are integral for the processes of leukocyte

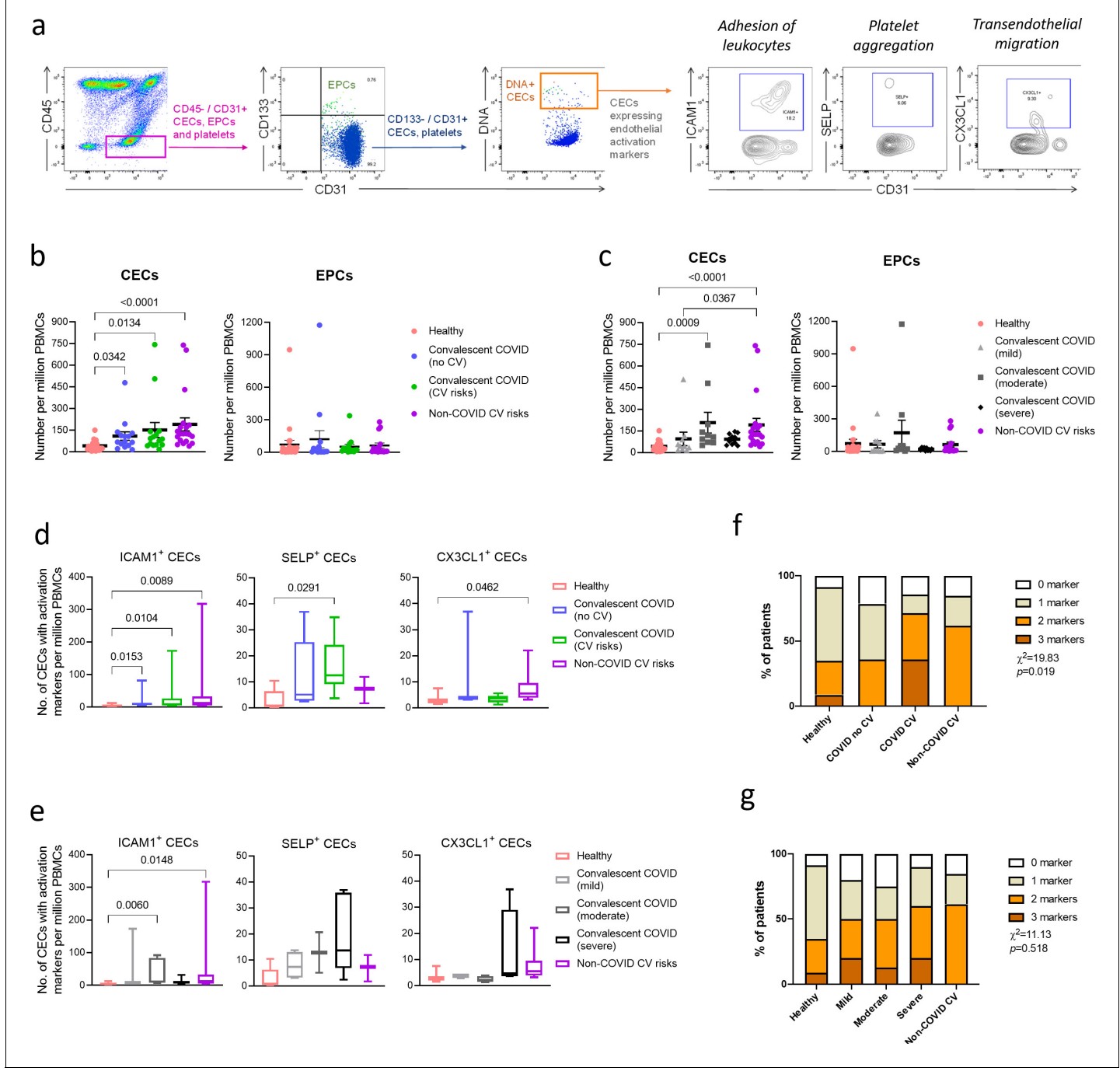

**Figure 1.** Enumeration and characterization of circulating endothelial cells (CECs) from COVID-19 convalescents (n = 30), non-COVID-19 patients (n = 20), and healthy participants (n = 24). (a) Using flow cytometry, CEC populations were gated from peripheral blood mononuclear cells (PBMCs) using a strategy involving positive (nuclear stain and CD31) and negative (CD45 and CD133) markers before characterization with three separate markers of endothelial activation. (b) and (c) Scatterplot visualization of the number of CECs and endothelial progenitor cells (EPCs) per million PBMCs identified from each sample with mean and standard error of mean for each group shown. (d) and (e) Boxplots extending from 25th to 75th percentile with bar showing mean and whiskers indicating the minimum and maximum number of CECs per million PBMCs from each group staining positive for endothelial activation markers ICAM1, SELP, or CX3CL1. Kruskal—Wallis test was performed (b—e) to test for difference between the groups with Dunn's multiple comparison test carried out for pairwise testing post hoc. (f) and (g) Cumulative analysis of patient frequency data of CECs staining positive for endothelial activation markers, ICAM1, SELP, and/or CX3CL1. Chi-squared goodness-of-fit test was performed to assess for difference in frequencies between the groups in (f) and (g).

adhesion, platelet aggregation, and trans-endothelial migration, respectively (*Goncharov et al., 2017*; *Johnson and Jackson, 2013*).

We enumerated and compared the mean CEC and EPC numbers per million PBMCs between healthy participants, convalescent COVID-19 patients, convalescent COVID-19 patients with cardiovascular risks, and patients with cardiovascular risk but no history of COVID-19. The mean CEC counts were significantly increased in all the convalescent COVID-19 groups and non-COVID-19 cardiovascular risk group compared to the healthy participants, while no significant difference was noted for EPCs (*Figure 1b*). Between the two convalescent COVID-19 groups, patients with cardiovascular risk had higher mean CEC numbers (M = 153.75 ± 51.55) than patients without cardiovascular risk (M = 116.21 ± 28.17). These observations are consistent with our hypothesis that COVID-19 patients, especially those with pre-existing cardiovascular risk, may display persistent signs of vascular dysfunction even after recovery from COVID-19.

To ascertain the consequence of COVID-19 on endothelial dysfunction, we regrouped the convalescent COVID-19 patients by their disease severity during acute infection (*Figure 1c*). We found that both convalescent COVID-19 patients who recovered from moderate symptoms (p=0.0009) and non-COVID-19 patients with cardiovascular risks (p<0.0001) had significantly higher numbers of CECs than healthy controls. It was surprising that convalescent COVID-19 patients from severe symptoms did not result in significantly higher CEC counts, probably due to limited representations in our current sample pool. With regard to EPC counts, no significant difference was observed possibly due to the absence of recent adverse vascular events in our groups with cardiovascular risks. EPC levels are more likely to elevate immediately following thrombotic events when EPCs are mobilized from the bone marrow into the bloodstream to facilitate vascular repair (*Massa et al., 2005*; *Shintani et al., 2001*; *Valgimigli et al., 2004*; *Yip et al., 2008*). Here, measurement of CECs seemed to be able to detect subclinical endothelial dysfunction. Collectively, *Figure 1b and c* demonstrate the potential impact of COVID-19 and cardiovascular risk factors on endothelial injury, as indicated by the elevated number of CECs in circulation.

We further performed molecular profiling of CECs for endothelial activation markers and used healthy participants as baseline. Our data revealed a significantly higher mean numbers of ICAM1$^+$ CECs in convalescent COVID-19 with or without cardiovascular risk and non-COVID-19 cardiovascular risk groups (*Figure 1d*). Only convalescent COVID-19 patients with cardiovascular risk had a higher SELP$^+$ CEC count (p=0.0291), while a higher CX3CL1$^+$ CEC count was found only in non-COVID-19 cardiovascular risk patients (p=0.0462). When we analyzed their endothelial activation hallmarks by disease severity (*Figure 1e*), both convalescent COVID-19 patients who recovered from moderate symptoms (p=0.006) and non-COVID-19 patients with cardiovascular risks (p=0.0148) had significantly higher numbers of ICAM1$^+$ CECs than healthy controls. We also observed a trend that convalescent COVID-19 patients who recovered from severe symptoms had the highest number of SELP$^+$ CECs, although insignificant. As seen in the spread of data in *Figure 1d and e*, large inter-individual variations in CEC characteristics were expected and could obscure observations of statistically significant comparisons. To overcome this, we compared the proportion of patients with regard to the numbers of activation markers expressed by CECs in each group, and a Chi-squared goodness-of-fit test was carried out to assess if the percentages of patients were similar across the groups. We found a significant relationship between the number of endothelial activation markers expressed by CECs and the disease status of patients, $X^2$ (9, *N* = 64)=19.83, p=0.019 where convalescent COVID-19 patients compounded with cardiovascular risk had the most pronounced endothelial activation hallmarks (*Figure 1f*), more than those having either a history of COVID-19 alone or cardiovascular risk without COVID-19. Conversely, COVID-19 severity did not render significantly different extent of endothelial activation hallmarks (*Figure 1g*). These findings demonstrate that COVID-19 may act in concert with comorbidities of hypertension and diabetes to intensify risks of future vascular complications due to elevated endothelial activation state.

## Persistent cytokine production in convalescent COVID-19 patients

Multiplex microbead-based immunoassay was performed to determine the cytokine levels in COVID-19 patient plasma during hospital admission and after discharge. To investigate the association between cytokine responses and cardiovascular risks, cytokine and chemokine levels were compared between COVID-19 patients with and without cardiovascular risk during the acute and convalescent phases of infection (*Figure 2a*). We found that COVID-19 patients with cardiovascular risk had lower

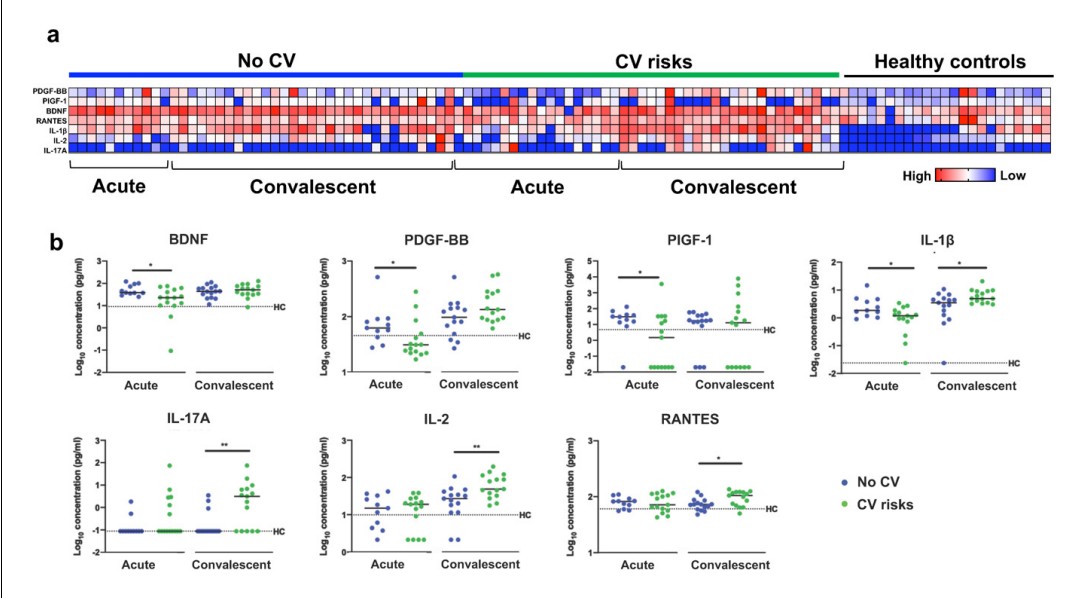

**Figure 2.** Plasma cytokine levels of COVID-19 patients with and without cardiovascular risk factors during the acute and convalescent phases of infection. Concentrations of 45 immune mediators were quantified using a 45-plex microbead-based immunoassay. (a) Heatmap of immune mediator levels in plasma samples of patients with (n = 15) and without (n = 15) cardiovascular risk factors at both acute (median 10 days post-illness onset), convalescent (0—90 days post-hospital discharge) phases of SARS-CoV-2 infection, and non-COVID-19 healthy controls. Each color represents the relative concentration of a particular analyte. Blue and red indicate low and high concentrations, respectively. (b) Profiles of significant immune mediators of COVID-19 patients with and without cardiovascular risk factors during acute and convalescent phases are illustrated as scatter plots. Cytokine levels in plasma fraction samples from first collection time point during hospital admission (acute, median 10 days post-illness onset) and discharge (convalescent, median 7 days post-hospital discharge) were compared. Mann—Whitney $U$ tests were performed on the logarithmically transformed concentration (*p<0.05; **p<0.01). Immune mediator levels for healthy control (n = 23) are indicated by the black dotted line. Patient samples with concentration out of measurement range are presented as the value of logarithm transformation of Limit of Quantification.

levels of growth factors brain-derived neurotrophic factor (median concentration 22.70 vs. 38.57 pg/mL, p=0.015), PDGF-BB (median concentration 31.08 vs. 62.51 pg/mL, p=0.023) and PIGF-1 (median concentration 30.69 vs. 1.49 pg/mL, p=0.040) compared to patients without cardiovascular risk at the early acute phase of infection (first plasma sample collected after hospital admission, at a median 10 days after symptom onset, IQR 7–15) (*Figure 2b*). These growth factors are known to promote vascular function (*Alomari et al., 2015*; *Brown et al., 1995*; *Carmeliet et al., 2001*), and our observations suggest that the endothelial dysfunction underlying patients with cardiovascular risk may impede vascular repair following injury during the acute phase of virus infection. Interestingly, the plasma levels of proinflammatory IL-1β (median concentration 4.95 vs. 3.51 pg/mL, p=0.026), IL-17A (median concentration 3.12 vs. 0.09 pg/mL, p=0.003), IL-2 (median concentration 48.82 vs. 27.07 pg/mL, p=0.007), and RANTES (median concentration 105.00 vs. 72.47 pg/mL, p=0.037) were significantly higher in patients with cardiovascular risk than those without (*Figure 2b*) during the early convalescent phase (first plasma sample collected after hospital discharge, median 7 days post hospital discharge, IQR 3–12). Notably, the levels proinflammatory cytokines such as IL-1β, IL-17A, IL-2, and RANTES remained elevated in COVID-19 patients at the early convalescent phase, particularly in those with prior cardiovascular risk, compared to healthy controls (*Figure 2b*). Result from a simple linear regression analysis showed that age was significantly associated with the presence of cardiovascular risks in our COVID-19 patients. After adjustment for age by multiple linear regression analysis, levels of RANTES at the early convalescent phase was significantly associated with cardiovascular risks (*Supplementary file 1*).

## Correlative studies implicating endothelial dysfunction with persistent cytokine production

To understand whether persistent immune activation may impact endothelial dysfunction, we performed correlation analysis of cytokine levels with the aforementioned CEC attributes. Interestingly,

CEC attributes from convalescent patients without prior cardiovascular risk correlated significantly with a greater number of cytokines than those with cardiovascular risk (*Figure 3a*). This may suggest that persistent cytokine production contributed primarily to endothelial injury (CEC counts) and activation hallmarks (CX3CL1+ and SELP+) in convalescent patients without previously known risk

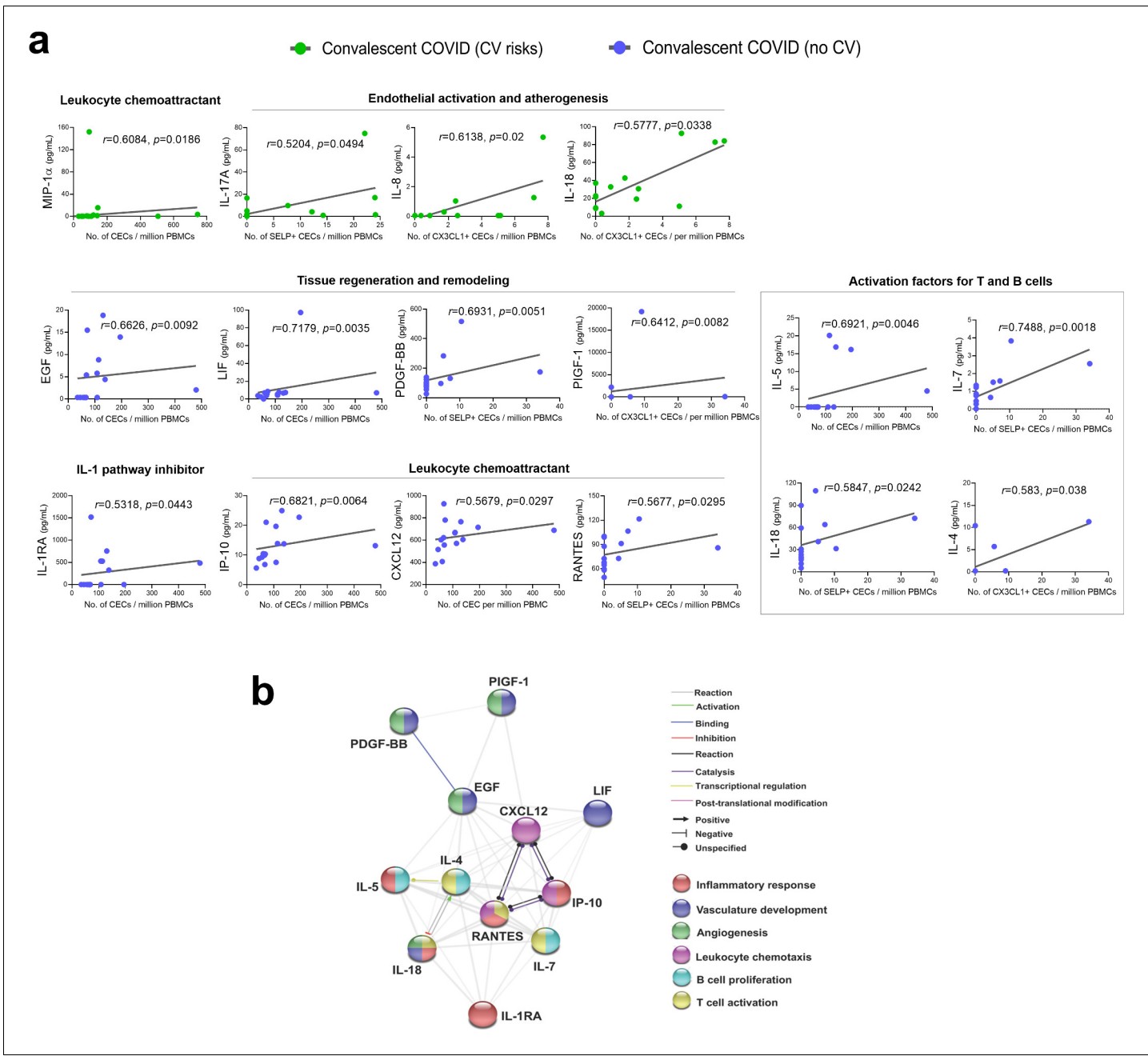

**Figure 3.** Significantly correlated cytokines with CEC attributes in COVID-19 convalescent patients. (a) Spearman rank correlation coefficients were calculated to assess the associations between the level of cytokines and CEC attributes in terms of mean numbers of CECs, SELP+ CECs, or CX3CL1+ CECs. Spearman's correlation coefficient r and p values (two-tailed test) were shown in plots. Source data relating to this figure is available. (b) Network analysis of CEC-associated cytokines in the convalescent COVID-19 patients without prior cardiovascular risk factors. Interactive relationships between the cytokines or chemokines were determined by Search Tool for the Retrieval of Interacting Genes/ Proteins (STRING) analysis, with a confidence threshold of 0.4.

The online version of this article includes the following source data for figure 3:

**Source data 1. Significantly correlated cytokines with CEC attributes in COVID-19 convalescent patients.**

factors. On the other hand, more pronounced endothelial injury and activation in those with cardio-vascular risk would have been attributed to their pre-existing cardiovascular risk factors, on top of the aftermath of COVID-19. Among the positively correlated cytokines in the cardiovascular risk group (*Figure 3a*), MIP-1α (CCL3) related to a chemoattractant for leukocytes, along with IL-17A (*Williams et al., 2019*), IL-8 (*Apostolakis et al., 2009*), and IL-18 (*Gerdes et al., 2002*), known to evoke activation of endothelial cells during atherogenesis, may suggest chronic development of atherosclerotic plaques already in these individuals.

For the convalescent patients without prior risk factors, the measures of CEC attributes reflected greater sensitivity to cytokine-driven endothelial dysfunction that could signify a more direct consequence of COVID-19. Further protein—protein interaction network analyses with Search Tool for the Retrieval of Interacting Genes/Proteins (STRING) highlighted the direct and indirect interactions between the significant cytokines and their functional associations with biological processes involved in inflammatory response, vasculature development, angiogenesis, leukocyte chemotaxis, B cell proliferation, and T cell activation (*Figure 3b*). Among the positively correlated cytokines (*Figure 3a*), we observed growth factors—associated tissue regeneration and remodeling, namely EGF, LIF, PDGF-BB, and PIGF-1 (PGF), indicating possibility of adaptive angiogenesis taking place as a response to preceding damages caused by viral infection and cytokine overdrive. IL-1RA, an IL-1 pathway inhibitor, could be an inflammation resolving mediator post infection. Conversely, IP-10 (CXCL10) is known to limit angiogenesis (*Bodnar et al., 2006*). This collective turnover of endothelial cells during blood vessel remodeling may result in elevated number of CECs. A number of significantly correlated cytokines are chemokines known to induce endothelial activation and promote chemotaxis of leukocytes to vascular endothelium. Chemotactic factors such as IP-10 (CXCL10), CXCL12, and RANTES (CCL5) regulate adhesion and transmigration of T lymphocytes, monocytes, and/or neutrophils through endothelial barrier (*Sokol and Luster, 2015*). Moreover, RANTES was specifically correlated with SELP$^+$ CEC (r = 0.5677, p=0.0295), indicating that activated platelet-derived RANTES may work in concert with endothelial SELP to mediate platelet aggregation and trigger coagulation cascade. In partial correlation analyses, a majority of these CEC—cytokines correlations were sustained even after accounting for age (*Supplementary file 1*).

Overall, CEC attributes of convalescent COVID-19 patients correlated with a vast majority of differentiation and activation factors associated with T cells and B cells (IL-4, IL-5, IL-7, IL-17A, IL-18, MIP-1α, and RANTES) (*Turner et al., 2014*), implicating a broad adaptive immune response with endothelial dysfunction. Besides being a target of the immune mediators, we recognize that endothelial cells, once activated, could also be a source of these cytokines, which in turn could activate their immune counterparts.

## Endothelial—immune crosstalk

Our ability to measure activation markers directly on CECs motivated us to better understand the receptor—receptor and chemokine—receptor interactions between activated endothelial cells with putative immune subpopulations in COVID-19. We performed data mining of published single-cell transcriptomic datasets on PBMCs from healthy participants, mild and severe COVID-19 patients (*Wilk et al., 2020*), and COVID-19 patients with or without cardiovascular disease (*Schulte-Schrepping et al., 2020*). Then, we re-analyzed the expressions of counter receptors (i.e., *ITGAL*, *SELPLG*, and *CX3CR1*) to our endothelial activation markers (i.e., *ICAM1*, *SELP*, and *CX3CL1*, respectively), in order to identify the potential immune interactors with activated endothelial cells. We found that those counter receptors were most pronouncedly expressed by CD8$^+$ T cells, natural killer (NK) cells, and to some extent monocytes (*Figure 4a* and *Figure 4—figure supplement 1*). In Wilk et al.'s dataset, the expression of *CX3CR1* was intensified in mild and severe COVID-19 patients than healthy participants (*Figure 4a*). The relationship between counter receptors expression and COVID-19 disease was even more marked in Schulte-Schrepping et al.'s dataset with a higher proportion of NK or both CD8$^+$ and NK cells expressing all three counter receptors found in mild and severe COVID-19 samples, respectively, regardless of comorbidity with cardiovascular disease (*Figure 4—figure supplement 1*). As the presence of those counter receptors generally marks cytotoxic effector lymphocytes in peripheral blood (*Nishimura et al., 2002*), we correspondingly found that CD8$^+$ T cells and NK cells were the major cytotoxic populations expressing perforin and granzymes (*Figure 4b*), especially in COVID-19 patients (*Figure 4—figure supplement 1*).

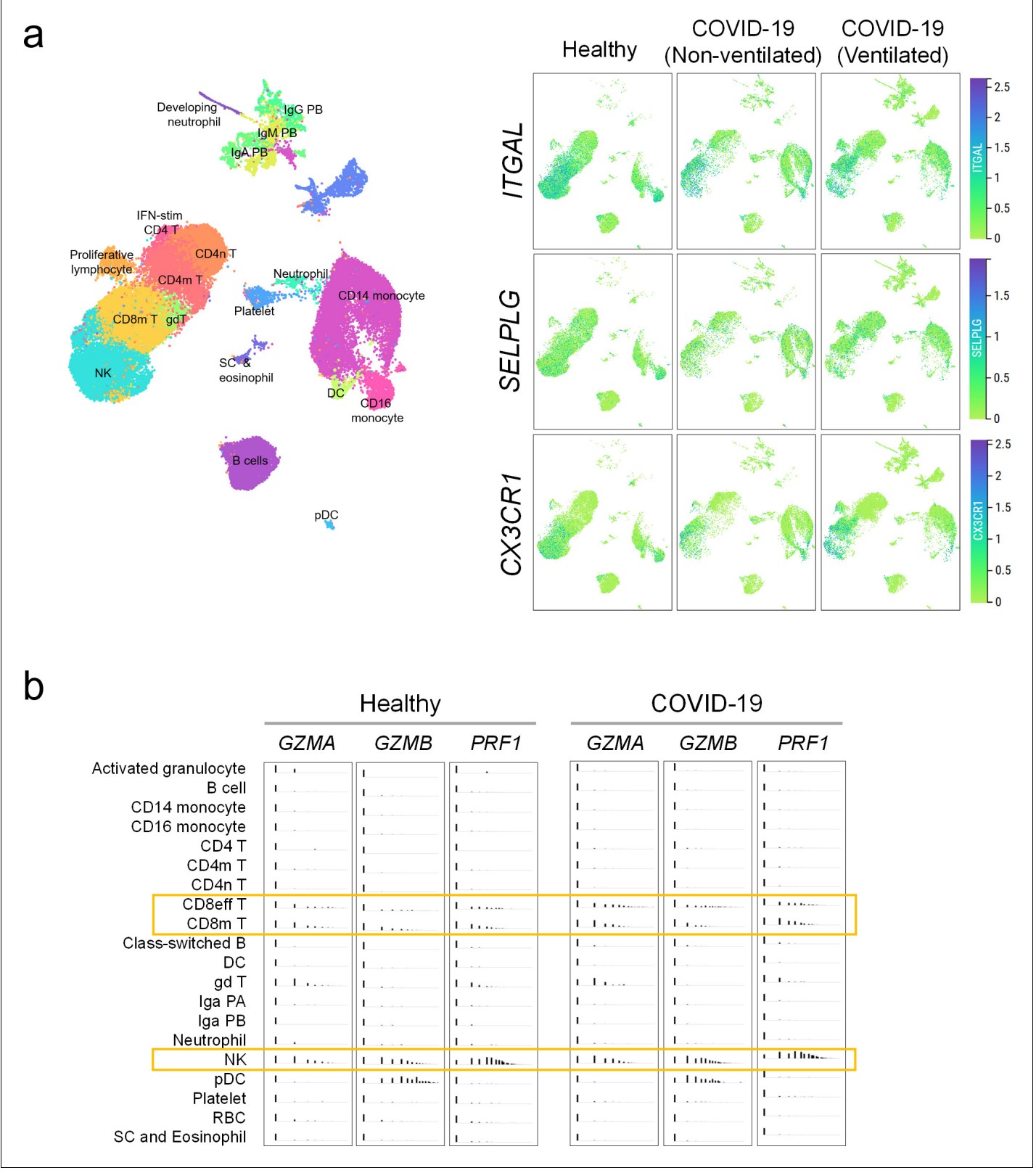

**Figure 4.** Immune interactors of activated endothelial cells. (**a**) UMAP representations of immune cell populations from healthy participants and COVID-19 patients annotated by cell types (left) and differential expressions of counter receptors *ITGAL*, *SELPLG*, and *CX3CR1*, which are known to interact with surface molecules of activated endothelial cells (right). (**b**) Distribution of the expressions of cytotoxic genes *GZMA*, *GZMB*, and *PRF1* across immune cell populations.

*Figure 4 continued*

The online version of this article includes the following figure supplement(s) for figure 4:

**Figure supplement 1.** COVID-19 samples with or without cardiovascular comorbidity have a higher proportion of immune cells expressing counter receptor and cytotoxicity-associated genes.

We hypothesize that persistent cytokine production activates endothelial cells in convalescent COVID-19 patient, which may promote interactions with cytotoxic effector cells. Therefore, we analyzed T cell features by immunophenotyping in our cohort (*Carissimo et al., 2020*) and found that convalescent COVID-19 patients, in particular those with underlying cardiovascular risk, had significantly higher frequencies of effector CD8$^+$ and CD4$^+$ T cells (EM), and central memory CD8$^+$ T cells (CD8 CM) than healthy controls (*Figure 5a*), while there were no notable differences across the groups for absolute numbers of circulating CD8$^+$/CD4$^+$ T cells. There is a possibility that individuals recovered from COVID-19 could be susceptible to vascular injury due to effector T cell cytotoxicity (*Figure 5b*). However, the results herein remain associative, and further experiments are required to establish cytotoxicity-induced endothelial injury in COVID-19 convalescents.

## Discussion

The rapid rise of COVID-19 'long-haulers' with lingering symptoms post-infection or recovered individuals who suffer from sudden cardiovascular events, would continue to place tremendous strain on our healthcare systems. About 2.5% of convalescent COVID-19 patients are found with thrombosis (including arterial and venous events), 30 days post discharge (*Patell et al., 2020*). As endothelial dysfunction often precedes serious thrombotic complications and ischemic damages to organs, we performed vascular phenotyping through analysis of CECs in convalescent COVID-19 patients. CECs are dysfunctional endothelial cells shed from damaged vessels, hence a surrogate marker of vascular injury (*Farinacci et al., 2019*; *Hill et al., 2003*). Our findings reveal that COVID-19 convalescents had significantly higher CEC count than non-COVID-19 healthy participants. Beyond enumeration of CECs, we further measured ICAM1, SELP, and CX3CL1 on CECs, expressions of which would suggest proinflammatory and procoagulant state of the endothelial cells originating from sites of vascular injury. In comparison with non-COVID-19 patients with cardiovascular risk (i.e., hypertension, diabetes, hyperlipidemia), the compounded effects of COVID-19 and prior cardiovascular risk factors rendered the most pronounced endothelial activation markers in COVID-19 convalescents. Activated endothelial cells are likely to release cytokines, which trigger extrinsic coagulation pathway (*Coccheri, 2020*), suggesting that recovered individuals may be susceptible to risk of thrombotic complications.

The sequence of events leading to post-COVID-19 complication has not been fully elucidated, but likely a vicious cycle caused by vascular leakage, activation of coagulation pathway, and inflammation (*Teuwen et al., 2020*). In this study, our cytokine profiling confirmed that cytokine production remained heightened post-infection. Of interest, the CEC attributes of our COVID-19 convalescents correlated significantly with a number of proinflammatory cytokines. We also found significantly higher proportions of effector CD8$^+$ and CD4$^+$ T cells in our COVID-19 convalescents, especially those with pre-existing cardiovascular risk, than healthy participants. Collectively, these suggest that prolonged overactive state of the immune system could be implicated with endothelial dysfunction. Published immunophenotyping studies on recovered COVID-19 patients are emerging. In many COVID-19 convalescents, broad CD8$^+$ T-cell response against multiple SARS-CoV-2 proteins mediated by effector and memory T cell populations were detected, suggesting a key role for T cells in protective immunity in recovered patients (*Dan et al., 2021*; *Grifoni et al., 2020*; *Oxford Immunology Network Covid-19 Response T cell Consortium et al., 2020*). It was reported that majority of these SARS-CoV-2 reactive T cells in COVID-19 convalescents had phenotypic markers of TEMRA (*Dan et al., 2021*) known to be more differentiated and cytotoxic in nature. Here, our analysis of published scRNA-seq datasets demonstrated that CD8$^+$ T cells in COVID-19 patients were associated with cytotoxic genes, as well as receptors that might potentially interact with counter receptors/ligands on activated endothelial cells. This led us to hypothesize that activated endothelial cells in COVID-19 convalescents could become a target of cytotoxic effector T

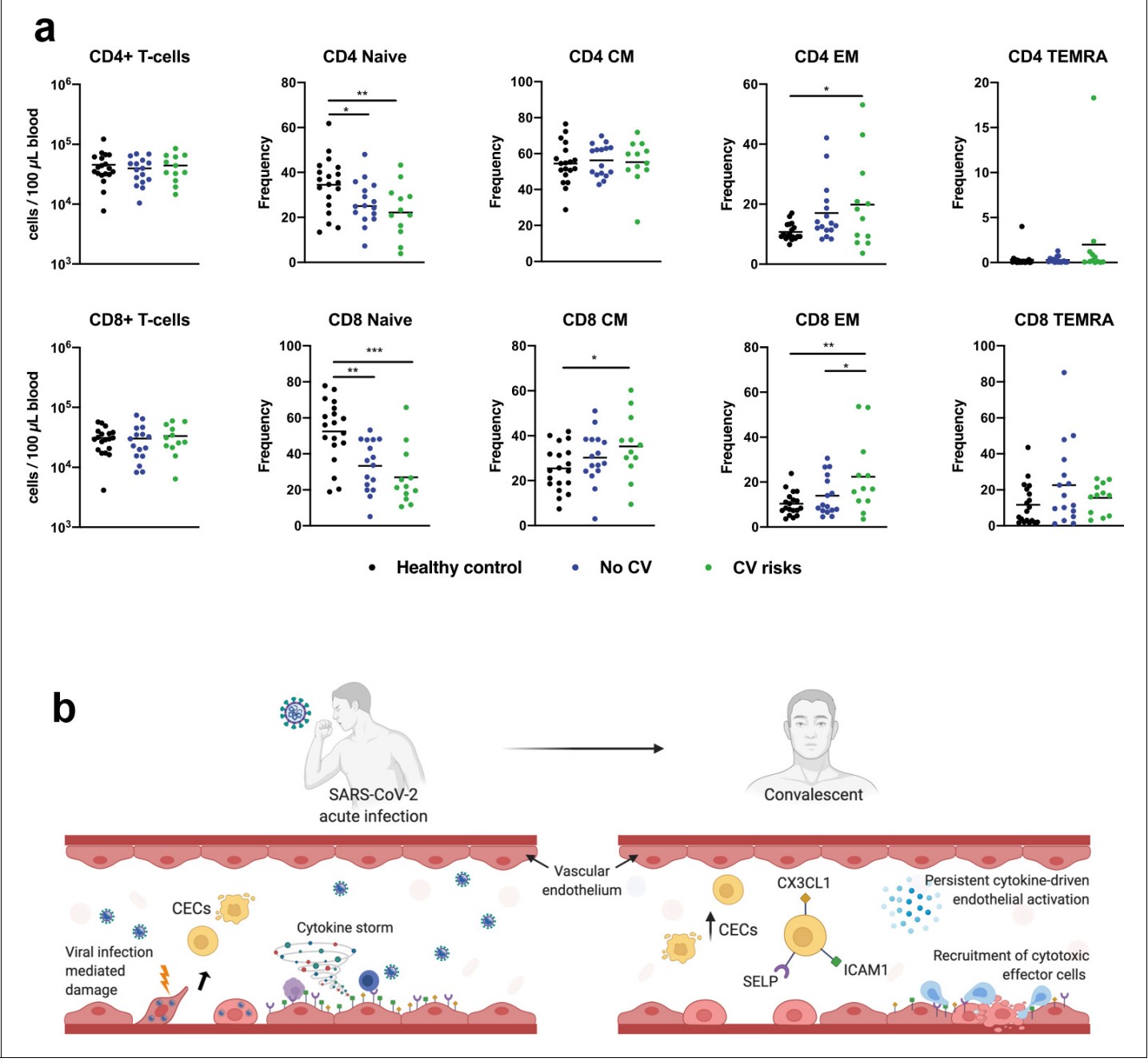

**Figure 5.** CD4[+] and CD8[+] T cells in COVID-19 patients with and without cardiovascular risk factors during the convalescent phase of infection. (a) Flow cytometry was performed on whole blood obtained from COVID-19 patients with (n = 12) and without (n = 16) cardiovascular risk factors at convalescent (11—50 days post hospital discharge) phase of SARS-CoV-2 infection, and non-COVID-19 healthy controls (n = 19). Naïve, terminally differentiated effector memory cells (TEMRA), central memory (CM), and effector memory (EM) T cells were characterized based on CD45RA and CD27 expressions. Absolute counts of CD4/CD8[+] T cells and individual frequencies of CD45RA vs. CD27 differentiation stage of T cells are illustrated as scatter plots. One-way ANOVA with post hoc tests were performed to compare the profiles across the groups (*p<0.05; **p<0.01; ***p<0.01). (b) Proposed mechanisms of endothelial dysfunction in COVID-19 from acute infection to convalescence.

lymphocytes. Endothelial injury could result in vascular permeability, further exposing tissue factor from underneath, which predisposes to risk of blood clotting (*van Hinsbergh, 2012*).

Immune cell-mediated endothelial injury has been observed in other viral infections, including Ebola, human cytomegalovirus infection, and malaria, where T cells recruited to infected site by

inflammation could induce apoptosis of infected endothelial cells, causing vascular leakage (*Claser et al., 2019*; *van de Berg et al., 2012*; *Wolf et al., 2015*; *Yang et al., 2000*). Likewise, in non-infectious diseases such as acute coronary syndrome and systemic sclerosis, cytotoxic CD4$^+$ T cells were capable of causing damages to endothelial cells (*Maehara et al., 2020*; *Nakajima et al., 2002*). It is also possible that some endothelial cells affected by SARS-CoV-2 infection had elicited pattern recognition receptors such as toll-like receptors, activating interferon pathway and inflammatory cytokine production that persist to convalescent phase. Antigen-experienced T cells preferentially adhere to activated endothelial cells and undergo transendothelial migration (*Carman and Martinelli, 2015*; *Marelli-Berg et al., 2004*), triggering vascular remodeling due to endothelial damage (*Cuttica et al., 2011*). Therefore, we postulate that activated and infected endothelial cells could be susceptible to direct cytotoxic T cell-mediated killing due to persistent immune activation in COVID-19 convalescents. However, in vitro experimentation with activated endothelial and immune cells is needed to confirm this hypothesis. How persistent cytokine response and effector T cell populations could translate into endothelial dysfunction during convalescence warrant further functional investigation.

We are mindful of the limitations with this current study. First, we noted our small sample size and other unmeasured confounding conditions, besides the current determinants of cardiovascular risks, which could contribute to some heterogeneity in our CEC and immune profiling. Second, convalescent blood samples were collected at various time points due to logistical constraints in recalling recovered patients on the same day post-infection. Finally, to understand post-COVID-19 complications, long-term phenotyping of vascular functions and immune profiles of convalescent patients is still lacking due to short-term horizons from when these individuals were first infected. There is a critical need to monitor prospective cohorts of recovered individuals in order to establish the full spectrum of clinical courses of complications. Global efforts in this aspect are evident and are underway.

Hematologic assessment (i.e., thrombin generation, platelet activation studies, von Willebrand factor) is commonly performed to help manage convalescent patients who have been re-admitted for acute thrombosis. It would be valuable to risk stratify convalescent patients before adverse events happen. Our profiling of CECs may serve as a form of vascular surveillance to complement hematologic assessment in identifying early molecular and endothelial changes in high-risk individuals. Analysis of CECs may detect subclinical changes in the vasculatures and be able to provide accurate and easily transferable endpoints in clinical assessment.

In summary, managing the aftermath of COVID-19 is an imperative. Endothelial instability may be a key mechanism underpinning the development of post-infection vascular complications. Clinical trials in preventive therapy for vascular complications may be needed.

# Materials and methods

## Key resources table

| Reagent type (species) or resource | Designation | Source or reference | Identifiers | Additional information |
|---|---|---|---|---|
| Biological sample (human) | Peripheral blood mononuclear cells | Singapore Immunology Network, National Heart Centre, Singapore | | Frozen PBMCs from healthy participants, convalescent COVID-19 patients, non-COVID-19 patients with cardiovascular risk |
| Antibody | Anti-mouse Ig, κ/negative control compensation particles set | BD Bioscience, San Jose, California, United States | Cat# 552843 RRID: AB_10051478 | To optimize fluorescence compensation settings for Flow Cytometry |
| Antibody | Hoechst 33342 Ready Flow Reagent | Invitrogen, Thermo Fisher Scientific, Waltham, Massachusetts, United States | Cat# R37165 | Flow Cytometry (5:100) |
| Antibody | PE/cyanine7 anti-human CD31 antibody (mouse IgG1, κ, clone# WM59) | Biolegend, San Diego, California, United States | Cat# 303118 RRID: AB_2247932 | Flow Cytometry (4:100) |

*Continued on next page*

*Continued*

| Reagent type (species) or resource | Designation | Source or reference | Identifiers | Additional information |
|---|---|---|---|---|
| Antibody | APC anti-human CD133 antibody (mouse IgG1, κ, clone# clone 7) | Biolegend, San Diego, California, United States | Cat# 372806 RRID: AB_2632882 | Flow Cytometry (4:100) |
| Antibody | Human CX3CL1/fractalkine chemokine domain Alexa Fluor 488-conjugated antibody (monoclonal mouse IgG1 clone# 51637) | Research And Diagnostic Systems, Inc. (R&D Systems, Inc.), Minneapolis, Minnesota, United States | Cat# IC365G-100UG RRID: AB_2885194 | Flow Cytometry (6:100) |
| Antibody | Alexa Fluor 488 anti-human CD62P (P-selectin) antibody (mouse IgG1, κ, clone# AK4) | Biolegend, San Diego, California, United States | Cat# 304916 RRID: AB_10719839 | Flow Cytometry (5:100) |
| Antibody | PE anti-human CD45 antibody (mouse IgG1, κ, clone# 2D1) | Biolegend, San Diego, California, United States | Cat# 368510 RRID: AB_2566370 | Flow Cytometry (3:100) |
| Antibody | BV711 mouse anti-human CD54 (mouse BALB/c IgG1, κ, clone# HA58 (RUO)) | BD Bioscience, San Jose, California, United States | Cat# 564078 RRID: AB_2738579 | Flow Cytometry (5:100) |
| Commercial assay or kit | Cytokine/ chemokine/ growth factor 45-Plex Human Procarta Plex Panel 1 | Invitrogen, Thermo Fisher Scientific, Waltham, Massachusetts, United States | Cat# EPX450-12171-901 | Luminex assay |
| Software, algorithm | Bio-Plex Manager 6.1.1 software | Bio-Rad Laboratories, Hercules, California, United States | | Data analysis of multiplex assay |
| Software, algorithm | GraphPad Prism, version 8.3.1 | GraphPad Software | | Data analysis, statistics, and graphing |
| Software, algorithm | FACSDiva software, version 8.0.1 | BD Bioscience, San Jose, California, United States | | Flow data acquisition |
| Software, algorithm | FlowJo software, version 10.7.1 | BD Bioscience, San Jose, California, United States | | Flow analysis |
| Software, algorithm | STRING (Search Tool for the Retrieval of Interacting Genes/Proteins) | ELIXIR Core Data Resources, Europe | | Protein-Protein interaction networks and functional enrichment analysis |
| Software, algorithm | Cellxgene | The Chan Zuckerberg Initiative Wilk et al dataset: https://cellxgene.cziscience.com /d/Single_cell_atlas_of_peripheral_ immune_response_to_SARS_ CoV_2_infection-25.cxg/ Schulte-Schrepping et al dataset: https://beta.fastgenomics.org/ datasets/detail-dataset-952687f71 ef34322a850553c4a24e82e#Cellxgene | | Analysis and visualization of scRNA-seq data |

## Study design, participants, and clinical data collection of convalescent COVID-19 patients

Convalescent COVID-19 individuals were recalled from the PROTECT study, which is a prospective observational cohort study at three public hospitals in Singapore (the National Centre for Infectious Diseases, National University Hospital, and Changi General Hospital). Written informed consent was obtained from participants who provided clinical data and biological samples. Study protocols were approved by ethics committees of the National Healthcare Group (2012/00917).

## Biological sample collection and processing for convalescent COVID-19 patients

The electronic medical records of patients enrolled in the PROTECT study were reviewed and their data entered onto a standardized collection form adapted from the International Severe Acute Respiratory and Emerging Infection Consortium's case record form for emerging severe acute respiratory infections. Serial blood samples were collected during hospitalization and post-discharge. Blood samples were processed as previously reported (*Singapore 2019 Novel Coronavirus Outbreak Research Team et al., 2020b*; *Young et al., 2020c*). PBMCs were isolated from whole blood collected in Cell Preparation Tubes (BD, #362761) for downstream CEC characterization. Plasma samples were separately stored for subsequent cytokine profiling.

## Study design, participants, and clinical data collection of healthy participants and non-COVID-19 patients with cardiovascular risk factors

Healthy participants and non-COVID-19 patients with cardiovascular risk factors were obtained from the Cardiac Ageing Study (*Koh et al., 2018*), which is a prospective observational cohort study performed at the National Heart Centre Singapore. The current study sample consisted of healthy participants who had no known cardiovascular disease or cerebrovascular disease or cancer. We included non-COVID-19 patients with cardiovascular risk factors for this comparison. All participants were examined and interviewed on one study visit by trained study coordinators. Participants completed a standardized questionnaire that included medical history and coronary risk factors. Sinus rhythm status was ascertained by resting electrocardiogram. Clinical data were obtained on the same day as biological sample collection.

## Biological sample collection and processing for healthy participants and non-COVID-19 patients with cardiovascular risk factors

Antecubital venous blood samples were taken from participants on the same day. After collection, the blood samples were immediately placed on ice for transportation and were processed within 6 hr to obtain buffy coat samples, which were subsequently cryopreserved.

## Vascular phenotyping by flow cytometry analysis

Study protocols were approved by ethics committees of the Nanyang Technological University Singapore (IRB-2020-09-011). PBMC samples were washed with Dulbecco's phosphate-buffered saline (DPBS; Hyclone, SH30028.02) + 1% BSA (Bovine Serum Albumin, Hyclone, SH30574.02), and then resuspended in 100 μL of DPBS + 1% BSA for antibody staining (Antibody panel in this section). Staining was carried out in the dark for 10 min at room temperature, followed by 20 min at 4°C on an analog tube rotator. After staining, cells were rinsed and resuspended in DPBS + 1% BSA for downstream flow cytometry analysis. Flow cytometry was performed using BD LSRFortessa X-20 (BD Bioscience, San Jose, California, United States ) and data acquisition was performed on FACSDiva software, version 8.0.1 (BD Bioscience, San Jose, California, United States ). Spectral overlap between INDO-1, APC, PE, PE-Cy7, AF488, and BV711 channels was calculated automatically by the FACSDiva software after measuring single-color compensation controls from pooled PBMCs. Optimal compensation was achieved using compensation control beads (anti-mouse Ig, κ/negative control compensation particles set, 552843) together with corresponding conjugated antibodies. Acquired data were analyzed using FlowJo software, version 10.7.1. Analysis of each patient typically included between 50,000 and 2,00,000 PBMCs depending on sample availabilities. CECs were detected by a combined immunophenotypic profile of $CD45^-$/$CD31^+$/$CD133^-$/$DNA^+$ and were further characterized for the expressions of ICAM1, SELP, and CX3CL1. Attributes such as CECs or "activated" CECs were represented as cells per million of PBMCs in *Figure 1*.

| Antibodies | Cat. number (Manufacturer) |
|---|---|
| Hoechst 33342 Ready Flow Reagent | R37165 (Invitrogen, Thermo Fisher Scientific, Waltham, Massachusetts, United States) |

*Continued on next page*

*Continued*

| Antibodies | Cat. number (Manufacturer) |
|---|---|
| APC-conjugated monoclonal antibody against human CD133 | 372806 (Biolegend, San Diego, California, United States ) |
| PE-conjugated monoclonal antibody against human CD45 | 304008 (Biolegend, San Diego, California, United States ) |
| PE-Cy7-conjugated monoclonal antibody against human CD31 | 303118 (Biolegend, San Diego, California, United States ) |
| AF488-conjugated monoclonal antibody against human CX3CL1/fractalkine chemokine domain | IC365G-100UG (Research And Diagnostic Systems, Inc. (R&D Systems, Inc.), Minneapolis, Minnesota, United States |
| BV711-conjugated monoclonal antibody against human C54 | 564078 (BD Bioscience, San Jose, California, United States) |
| AF488-conjugated monoclonal antibody against human CD62P | 304916 (Biolegend, San Diego, California, United States ) |

## Data mining of published immune single-cell transcriptomes of COVID-19

Single cell transcriptomic datasets published by *Wilk et al., 2020* (*Figure 4*) were re-analyzed using the cellxgene platform (https://cellxgene.cziscience.com/d/Single_cell_atlas_of_peripheral_immune_response_to_SARS_CoV_2_infection-25.cxg/). Briefly, the uniform manifold approximation and projection (UMAP) visualization was annotated using the metadata included using the "cell_type_fine" taxonomy. The expression data were obtained by interrogating the individual expression of the genes in the UMAP (*CX3CR1, ITGAL, and SELPLG*) or across the various immune population included in the "cell_type_fine" taxonomy (*PRF1, GZMA*, and *GZMB*). The single cell datasets published by *Schulte-Schrepping et al., 2020* were re-analyzed as described in the *Figure 4—figure supplement 1*.

## Cytokine analysis by multiplex microbead-based immunoassay

Plasma samples were treated with 1% Triton X-100 solvent-detergent mix for virus inactivation (*Darnell and Taylor, 2006*). Cytokine levels in COVID-19 patient plasma across different acute and convalescent time points were measured with the Luminex assay using the cytokine/chemokine/growth factor 45-plex Human ProcartaPlex Panel 1 (ThermoFisher Scientific). The Cytokine/Chemokine/Growth Factor 45-plex Human ProcartaPlex Panelone panel included granulocyte-macrophage colony-stimulating factor (GM-CSF), epidermal growth factor (EGF), brain-derived neurotropic factor, beta-nerve growth factor (bNGF), basic fibroblast growth factor (FGF-2), hepatocyte growth factor (HGF), monocyte chemoattractant protein (MCP) 1, macrophage inflammatory protein (MIP) 1α, MIP-1β, RANTES (regulated on activation, normal T cell expressed and secreted), chemokine (C-X-C motif) ligand (CXCL) 1 (GRO-α), stromal cell-derived factor 1 (SDF-1α), interferon (IFN) gamma-induced protein 10 (IP-10), eotaxin, IFN-α, IFN-γ, interleukin (IL) IL-1α, IL-1β, IL-1RA, IL-2, IL-4, IL-5, IL-6, IL-7, IL-8, IL-9, IL-10, IL-12p70, IL-13, IL-15, IL-17A, IL-18, IL-21, IL-22, IL-23, IL-27, IL-31, leukemia inhibitory factor (LIF), stem cell factor (SCF), tumor necrosis factor (TNF)-α and -β, vascular endothelial growth factors A and D (VEGF-A, VEGF-D), platelet derived growth factor (PDGF-BB), and placental growth factor (PLGF-1). Standards and plasma from COVID-19 patients and healthy controls were incubated with fluorescent-coded magnetic beads pre-coated with respective antibodies in a black 96-well clear-bottom plate overnight at 4°C. After incubation, plates were washed five times with wash buffer (PBS with 1% BSA [Capricorn Scientific GmbH, Ebsdorfergrund, Germany]) and 0.01% Tween (Promega Corporation, Madison, Wisconsin, United States). Sample—antibody—bead complexes were incubated with Biotinylated detection antibodies for 1 hr and washed five times with wash buffer. Subsequently, Streptavidin-PE was added and incubated for another 30 min. Plates were washed five times again before sample—antibody—bead complexes were re-suspended in sheath fluid for acquisition on the FLEXMAP 3D (Luminex) using xPONENT 4.0 (Luminex) software. Internal control samples were included in each Luminex assays to remove any potential plate effects. Readouts of these samples were then used to normalize the assayed plates. A correction factor was

obtained from the differences observed across the multiple assays and this correction factor was then used to normalize all the samples. Standard curves were generated with a 5-PL (5-parameter logistic) algorithm, reporting values for mean florescence intensity (MFI) and concentration data. The concentrations were logarithmically transformed to ensure normality. Patient samples with a concentration out of measurement range were assigned the value of the logarithmic transformation of the limit of quantification. Data analysis was done with Bio-Plex Manager 6.1.1 software.

## T cells phenotyping by flow cytometry analysis

Whole blood was stained with antibodies for 20 min in the dark at room temperature, as reported previously (*Carissimo et al., 2020*). Samples were then supplemented with 0.5 mL of 1.2× BD FACS lysing solution (BD 349202). Final 1× concentration taking into account volume in tube before addition is ~1% formaldehyde to fix viruses as well as lyse the red blood cells. Samples were vortexed and incubated for 10 min at room temperature. About 500 µL of PBS (Gibco, #10010–031) was added to wash the samples and spun at 300 *g* for 5 min. Washing step of samples were repeated with 1 mL of PBS. Samples were then transferred to polystyrene FACS tubes containing 10 µL (10,800 beads) of CountBright Absolute Counting Beads (Invitrogen, Thermo Fisher Scientific, Waltham, Massachusetts, United States, #36950). Samples were then acquired using BD LSRII five laser configuration using automatic compensations. For analysis of flow cytometry data, FlowJo version 10.6.1 was used for analysis of flow cytometry data. T cell populations were identified by previously reported gating strategies (*Carissimo et al., 2020*).

## Statistics

Due to inter-individual heterogeneities of flow cytometry and cytokine data, nonparametric tests of association were preferentially used throughout this study unless otherwise stated. For flow cytometry data on CEC attributes, statistical differences between groups were calculated using Kruskal—Wallis test with Dunn's multiple comparison post-test. Significant p values ($<0.05$) were indicated on the graphs directly. To assess for differences in the frequencies of patients with CECs expressing endothelial activation markers, ICAM1, SELP, or CX3CL1 and the cumulative frequencies for all markers, the number of patients per group were summarized in contingency tables. These were then analyzed with Chi-squared goodness-of-fit test with expected values generated from the data using GraphPad Prism, version 8.3.1.

For the cytokine analysis, Mann—Whitney *U* tests were used to discern the differences in cytokine levels between the patients with or without cardiovascular risk factors. A multiple linear regression analysis was conducted to examine the association between plasma cytokines and the presence of cardiovascular risks in COVID-19 patients after adjustment for age. Heatmap and scatter plots were generated using GraphPad Prism version 8. Concentrations of immune mediators were scaled between 0 and 1 for visualization in the heatmap. Protein—protein interaction networks of the CEC-associated cytokines were predicted and illustrated with Search Tool for the Retrieval of Interacting Genes/Proteins database (STRING) (version 11.0; available at: https://string-db.org). All the interactions between cytokines were derived from high-throughput laboratory experiments and previous knowledge in curated databases at a confidence threshold of 0.4.

For correlative study between CEC attributes and cytokine levels, we selected non-parametric Spearman correlation with no assumption regarding value distribution. Non-linear regression and Spearman's rank correlation coefficients were calculated to access associations between the level of serum cytokines and CEC attributes for convalescent COVID-19 patients with and without cardiovascular risk factors. The level of statistical significance was set at two-tailed test with p values $<0.05$ to detect any significant associations between specific cytokines and CEC attributes. Statistical analyses for this correlative study were performed using GraphPad Prism software, version 8.3.1. To address potential age-related contributions to the correlations observed between CEC and plasma cytokines in *Figure 3*, non-parametric partial correlation coefficients based on Spearman's rank correlations were determined for the identified correlated variables while controlling for age. The R package 'ppcor' was used to compute the coefficients, p values and test statistics in Supplementary File. (*Kim, 2015*).

**Study approval**

This study was approved by the Local Ethics Committee of the National Healthcare Group (2012/00917) and Nanyang Technological University Singapore Institutional Review Board (IRB-2020-09-011).

## Acknowledgements

We thank all clinical and nursing staff who provided care for the patients; staff in the Singapore Infectious Disease Clinical Research Network and Infectious Disease Research and Training Office of the National Centre for Infectious Diseases for coordinating patient recruitment. We also wish to thank Ms Siti Naqiah Amrun, Ms Rhonda Sin-Ling Chee, Mr Nicholas Kim-Wah Yeo, Mr Anthony Torres-Ruesta, Drs Chek Meng Poh, Cheryl Yi-Pin Lee, Matthew Tay, and Zi-Wei Chang from the Singapore Immunology Network (SIgN), for their help in isolating PBMCs and plasma fractions from the blood of COVID-19 patients. We are also grateful to Dr Danielle Anderson and her team at Duke-NUS, for their technical assistance in virus inactivation procedures with Triton-X-100; Dr Olaf Rötzschke, Dr Bernett Lee, Wilson How, and Norman Leo Fernandez from the Singapore Immunology Network (SIgN) Multiplex Analysis of Proteins (MAP) platform, for their assistance in running multiplex microbead-based immunoassay. We also thank Ms Manisha Cooray, Dr Chun-Yi Ng, and Ms Natalie Yeo for their technical help during experimental optimization.

## Additional information

### Competing interests

Barnaby E Young: Barnaby E. Young declares no direct competing interests with this work, but has received honorarium outside this work from Sanfoi and Roche. Paul A Tambyah: Paul A. Tambyah declares no direct competing interests with this work but has received research support outside this work from Roche, Sanofi-Pasteur, Johnson and Johnson, AJ Biologicals and Shionogi. The other authors declare that no competing interests exist.

### Funding

| Funder | Grant reference number | Author |
|---|---|---|
| National Medical Research Council | COVID19RF-001 | Siew-Wai Fong<br>Barnaby E Young<br>Yi-Hao Chan<br>Guillaume Carissimo<br>Seow-Yen Tan<br>Paul A Tambyah<br>Laurent Renia<br>Lisa FP Ng<br>David C Lye |
| National Medical Research Council | COVID19RF-060 | Siew-Wai Fong<br>Barnaby E Young<br>Yi-Hao Chan<br>Guillaume Carissimo<br>Seow-Yen Tan<br>Paul A Tambyah<br>Laurent Renia<br>Lisa FP Ng<br>David C Lye |
| Biomedical Research Council, A*STAR | H20/04/g1/006 | Siew-Wai Fong<br>Yi-Hao Chan<br>Guillaume Carissimo<br>Laurent Renia<br>Lisa FP Ng |
| National Research Foundation Singapore | NRF2017_SISFP09 | Siew-Wai Fong<br>Yi-Hao Chan<br>Guillaume Carissimo<br>Laurent Renia<br>Lisa FP Ng |

| Nanyang Technological University | Nanyang Assistant Professorship Start-Up Grant | Christine Cheung Anthony Siau Shuba Krishnan |
|---|---|---|
| Ministry of Education - Singapore | MOE2018-T2-1-042 | Christine Cheung Florence WJ Chioh Kan-Xing Wu |
| Agency for Science, Technology and Research | H18/01/a0/017 | Christine Cheung |
| National Medical Research Council | NMRC/TA/0031/2015 MOH-000153 NMRC/OFIRG/0018/2016 NMRC/BnB/0017/2015 MOH-000358 | Louis LY Teo Fei Gao Ru San Tan Liang Zhong Angela S Koh |

The funders had no role in study design, data collection and interpretation, or the decision to submit the work for publication.

## Author contributions

Florence WJ Chioh, Data curation, Formal analysis, Validation, Investigation, Visualization, Methodology, Writing - review and editing; Siew-Wai Fong, Data curation, Formal analysis, Investigation, Visualization, Methodology, Writing - original draft, Writing - review and editing; Barnaby E Young, Resources, Data curation, Formal analysis, Funding acquisition, Writing - review and editing; Kan-Xing Wu, Formal analysis, Methodology, Writing - original draft, Writing - review and editing; Anthony Siau, Formal analysis, Visualization, Writing - original draft, Writing - review and editing; Shuba Krishnan, Methodology; Yi-Hao Chan, Data curation, Formal analysis; Guillaume Carissimo, Data curation, Formal analysis, Methodology; Louis LY Teo, Fei Gao, Ru San Tan, Liang Zhong, Resources, Funding acquisition, Writing - review and editing; Angela S Koh, Resources, Data curation, Funding acquisition, Writing - review and editing; Seow-Yen Tan, Paul A Tambyah, Resources, Data curation; Laurent Renia, Conceptualization, Supervision, Funding acquisition, Writing - review and editing; Lisa FP Ng, David C Lye, Conceptualization, Resources, Supervision, Funding acquisition, Writing - review and editing; Christine Cheung, Conceptualization, Formal analysis, Supervision, Funding acquisition, Investigation, Visualization, Methodology, Writing - original draft, Project administration, Writing - review and editing

## Author ORCIDs

Yi-Hao Chan http://orcid.org/0000-0003-0329-238X
Seow-Yen Tan http://orcid.org/0000-0001-7870-2879
Laurent Renia http://orcid.org/0000-0003-0349-1557
Christine Cheung https://orcid.org/0000-0001-7127-9107

## Ethics

Human subjects: This study was approved by the Local Ethics Committee of the National Healthcare Group (2012/00917) and Nanyang Technological University Singapore Institutional Review Board (IRB-2020-09-011). Written informed consent was received from participants prior to inclusion in the PROTECT study.

## Decision letter and Author response

Decision letter https://doi.org/10.7554/eLife.64909.sa1
Author response https://doi.org/10.7554/eLife.64909.sa2

# Additional files

## Supplementary files

• Supplementary file 1.
• Transparent reporting form

## Data availability

All data generated or analysed during this study are included in the manuscript and supplemental files. Source data files have been provided for Figure 3.

The following previously published datasets were used:

| Author(s) | Year | Dataset title | Dataset URL | Database and Identifier |
|---|---|---|---|---|
| Wilk AJ, Rustagi A, Zhao NQ, Roque J, Martinez-Colon GJ, McKechnie JL, Ivison GT, Ranganath T, Vergara R, Hollis T, Simpson LJ, Grant P, Subramanian A, Rogers AJ, Blish CA | 2020 | A single-cell atlas of the peripheral immune response to severe COVID-19 | https://www.ncbi.nlm.nih.gov/geo/query/acc.cgi?acc=GSE150728 | NCBI Gene Expression Omnibus, GSE150728 |
| Schulte-Schrepping J, Reusch N, Paclik D, Bassler K, Schlickeiser S, Zhang B, Kramer B, Krammer T, Brumhard S, Bonaguro L, De Domenico E, Wendisch D, Grasshoff M, Kapellos TS, Beckstette M, Pecht T, Saglam A, Dietrich O, Mei HE, Schulz AR, Conrad C, Kunkel D, Vafadarnejad E, Xu CJ, Horne A, Herbert M, Drews A, Thibeault C, Pfeiffer M, Hippenstiel S, Hocke A, Muller-Redetzky H, Heim KM, Machleidt F, Uhrig A, Bosquillon de Jarcy L, Jurgens L, Stegemann M, Glosenkamp CR, Volk HD, Goffinet C, Landthaler M, Wyler E, Georg P, Schneider M, Dang-Heine C, Neuwinger N, Kappert K, Tauber R, Corman V, Raabe J, Kaiser KM, Vinh MT, Rieke G, Meisel C, Ulas T, Becker M, Geffers R, Witzenrath M, Drosten C, Suttorp N, von Kalle C, Kurth F, Handler K, Schultze JL, Aschenbrenner AC, Li Y, Nattermann J, Sawitzki B, Saliba AE, Sander LE, Deutsche Covid-Omics Initiative | 2020 | ScRNA-seq of PBMC and whole blood samples reveals a dysregulated myeloid cell compartment in severe COVID-19 | https://www.ebi.ac.uk/ega/studies/EGAS00001004571 | ebi, EGAS00001004571 |

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
