## [Decision Letter]

**Acceptance summary:**

This paper likely will be of broad interest to scientists studying vascular complications in convalescent coronavirus disease 2019 (COVID-19) patients. The authors present new insights into endothelial-immune cells crosstalk as an underlying mechanism of vascular complications observed in COVID-19 patients after initial recovery. The findings presented in this this paper will be of value to further understand and to develop new approaches that might prevent future vascular complication prevention in convalescent COVID-19 patients.

**Decision letter after peer review:**

Thank you for submitting your article "Convalescent COVID-19 patients are susceptible to endothelial dysfunction due to persistent immune activation" for consideration by *eLife*. Your article has been reviewed by three peer reviewers, one of whom is a member of our Board of Reviewing Editors, and the evaluation has been overseen by a Senior Editor. The following individuals involved in review of your submission have agreed to reveal their identity: Sergio Coccheri (Reviewer #2); Maximilian Ackermann (Reviewer #3).

We would like you to prepare a revised submission. Please take note changes in our policy on revisions we have made in response to COVID-19 (https://elifesciences.org/articles/57162). Specifically, when editors judge that a submitted work as a whole belongs in *eLife* but that some conclusions require a modest amount of additional new data, as they do with your paper, we are asking that the manuscript be revised to either limit claims to those supported by data in hand, or to explicitly state that the relevant conclusions require additional supporting data.

Summary:

This paper addresses a number of complications, mainly cardiovascular or renal, that may affect patients' convalescent from a COVID-19 episode. The authors suggest that an increased number of CECs as observed in blood of COVID convalescent subjects, is one of the main factors that are affected by pre-existing cardiovascular risk factors. The evaluation of CECs was followed by related measurements regarding activating circulating factors such as ICAM, P-selectin, cytokines and others released by CECs. The manuscript provides new insights regarding endothelial-immune cells crosstalk as an underlying mechanism contributing to vascular dysfunction in COVID-19 patients after initial recovery. The findings reported in this paper further our understanding of vascular complication in convalescent patients and provide the basis to develop novel preventive approaches for patients diagnosed with COVID-19.

Revisions:

There are several comments and suggestions:

1) The number of COVID patients recruited is limited (30 patients), considering that they are divided into light, moderate and severe patients and that the comparisons involved healthy subjects, patients' convalescent from COVID 19 with or without CV risk factors, and subjects with CV risk factors but no previous COVID. Moreover, the high number of Authors (18) and of Centers (13), should have enrolled more COVID convalescent patients. It is therefore difficult to apply statistical evaluations. For instance, insufficient n numbers could explain the fact that significance was obtained only between healthy subjects and non-COVID patients with risk factors, as already known. But other, more important comparisons could be significant with a higher n numbers. The number of CEC was also much higher in COVID convalescent patients with versus without CV risk factors.

2) The authors measured factors produced by activated endothelial cells such as ICAM, P-selectin, cytokines. They also hypothesized that activated endothelial cells may be attacked by CD8^+^T or natural killer cells. However, no direct evidence for such a mechanism is presented. This authors should provide some mechanistic evidence concerning the involvement of these factors in COVID patients.

3) How do the authors explain a possible functional role of elevated levels of circulating endothelial cells? Are they incorporated in the vasculature, and if so, how? There is evidence that circulating endothelial cells play a functional role for the vascular repair by intussusceptive angiogenesis in COVID-19 (PMID: 33008942, PMID: 32437596), but also in prolonged inflammatory responses, such as pulmonary fibrosis (PMID: 31806721). The authors also need to discuss the potential mechanistic implications of CEC for thrombotic events such as DVT, pulmonary embolism, and myocardial infarction as a frequent and often fatal complication in COVID patients, with reference to the corresponding literature.

4) It is surprising that the authors not include sorting for CD34+/CD133+ cells. These data should be added to the manuscripts, and the implications of the results should be discussed.

5) The authors could comment one or two sentences more on the interaction between T-cells and endothelial cells. What is the impact of toll-like receptors (TLRs) on the remodeling?

6) If possible, this manuscript will give us more mechanistic insight if the author could add some in vitro experiments using endothelial cells and immune cell lines infected with the COVID-19 virus.

---

## [Author Response]

Revisions:There are several comments and suggestions:1) The number of COVID patients recruited is limited (30 patients), considering that they are divided into light, moderate and severe patients and that the comparisons involved healthy subjects, patients' convalescent from COVID 19 with or without CV risk factors, and subjects with CV risk factors but no previous COVID. Moreover, the high number of Authors (18) and of Centers (13), should have enrolled more COVID convalescent patients. It is therefore difficult to apply statistical evaluations. For instance, insufficient n numbers could explain the fact that significance was obtained only between healthy subjects and non-COVID patients with risk factors, as already known. But other, more important comparisons could be significant with a higher n numbers. The number of CEC was also much higher in COVID convalescent patients with versus without CV risk factors.

We agree that low number of patients is a limitation of this study, which has initially rendered statistical differences insignificant for important comparisons. Therefore, we have increased the number of healthy participants (n=24) to make them more comparable to the size of our convalescent COVID patients (n=30). Indeed, statistical evaluations have improved and significant differences in CEC counts are achieved between healthy participants compared to convalescent COVID-19 subjects, and non-COVID-19 subjects with cardiovascular risk (Figure 1B and 1C). Likewise for CECs with activation marker, statistical significance is achieved mainly for ICAM1+ CECs (Figure 1D and 1E). We have also stated that small sample size remains a limitation of this study in Discussion section.

2) The authors measured factors produced by activated endothelial cells such as ICAM, P-selectin, cytokines. They also hypothesized that activated endothelial cells may be attacked by CD8^+^T or natural killer cells. However, no direct evidence for such a mechanism is presented. This authors should provide some mechanistic evidence concerning the involvement of these factors in COVID patients.

We further analysed T cell data obtained from our immunophenotyping study of the same cohort and found that COVID-19 patients with cardiovascular risks had higher frequency of effector CD8^+^ and CD4^+^ T cells at the convalescent phase of infection. The new results are now added as Figure 5A. We have moderated our claims, recognizing that the results herein remain associative and further experiments are required to establish cytotoxicity-induced endothelial injury in COVID-19 convalescents.

3) How do the authors explain a possible functional role of elevated levels of circulating endothelial cells? Are they incorporated in the vasculature, and if so, how? There is evidence that circulating endothelial cells play a functional role for the vascular repair by intussusceptive angiogenesis in COVID-19 (PMID: 33008942, PMID: 32437596), but also in prolonged inflammatory responses, such as pulmonary fibrosis (PMID: 31806721). The authors also need to discuss the potential mechanistic implications of CEC for thrombotic events such as DVT, pulmonary embolism, and myocardial infarction as a frequent and often fatal complication in COVID patients, with reference to the corresponding literature.

Please allow us to clarify two major types of blood endothelial cells and their origins. (1) CECs are detached from the vessel endothelium as a result of injury and/or disease (PMID: 28459428). On the other hand, (2) circulating EPCs originate from the bone marrow and are mobilized into the bloodstream in response to vascular injury. Existing literature has not reported functional role on CECs. Blood borne CECs originate from *in situ* damaged endothelial cells and are dysfunctional in nature, hence unlikely to be re-incorporated into vasculatures. Our manuscript involves the study of CECs as an indicator of endothelial dysfunction.

As the reviewer has aptly pointed out about incorporation into vasculatures, we believe this context may relate to the well-studied role of circulating EPCs. In both COVID-19 and interstitial lung disease studies (PMIDs: 33008942, 32437596, 31806721), the research team found enhanced sprouting and intussusceptive angiogenesis. These phenomena are associated with the recruitment of circulating EPCs, in line with the increased EPC levels observed in COVID patients (PMID: 32762140), in supporting new vessel formation. EPCs are known to take part in endothelial repair (PMID: 26187355), prompting the development of multiple cardiovascular therapeutic strategies (PMID: 30619864).

In contrast to EPCs which arise from the bone marrow and have a known role in vascular repair, the CECs studied in our manuscript are mature and dysfunctional endothelial cells shed from damaged blood vessels, hence a direct surrogate to characterize the state of vascular health in convalescent COVID-19 patients. Besides enumeration of CECs, we further evaluated endothelial activation markers such as ICAM-1 and P-selectin (Figure 1D and 1E). The profiling of such activation markers is possible in CECs and may reflect retrospectively the state of endothelial health, just before they were detached from damaged vessels. More pronounced activation hallmarks, as seen in convalescent COVID-19 patients with underlying cardiovascular risk, may imply their activated endothelial cells’ enhanced cytokine release which could trigger extrinsic coagulation pathways, leading to risk of thrombotic complications (PMID: 33090354).

To improve clarity of our manuscript, we have better defined the biological context of CECs versus EPCs, citing the relevant references. Also, the implication of CECs (endothelial damage) to risk of thrombosis is added.

4) It is surprising that the authors not include sorting for CD34+/CD133+ cells. These data should be added to the manuscripts, and the implications of the results should be discussed.

Thank you reviewer for your suggestion. The sorting for CD34+/CD133+ cells refers more specifically to circulating EPCs. We agree that EPCs should also constitute a valid biomarker of vascular injury. With reference to our response to point 3 above, due to the different origins of CECs and EPCs, we have chosen to measure CECs (CD45-/CD31+/CD133-/DNA+) as our current aim is to elucidate the extent of endothelial dysfunction in convalescent COVID-19 patients. CECs which are shed from damaged vessels could be directly reflective of *in situ* endothelial pathophysiology. Furthermore, the study of CECs would enable the profiling of endothelial activation markers.

For completeness of our manuscript, we have added new data on CD133+/CD45-/DNA^+^ population which contains putative circulating EPCs (PMIDs: 12778169, 15226175, 32762140) in Figures 1B and 1C. There were no significant differences in EPC counts across healthy participants, COVID-19 convalescent patients and non-COVID-19 subjects with cardiovascular risk. We are mindful that increased number of patients and the use of additional marker such as CD34 would improve robustness of identifying EPCs. Another reason for non-significant differences may akin to inconsistency with increased or decreased levels of EPCs observed in cardiovascular diseases at various stages of progression (PMID: 25984328). By virtue of the nature of EPCs, we may expect alterations of EPC levels most pronouncedly following thrombotic events when EPCs are recruited from bone marrow, into the circulation, and finally to sites of vascular ischemia to carry out vascular repair. In our current cohorts, we analyzed COVID-19 convalescent patients and non-COVID-19 subjects with cardiovascular risk but in the absence of adverse vascular events, thus may partly explain the lack of statistical significance. Nonetheless, measurement of CECs seem to be able to detect subclinical endothelial dysfunction by increased CEC counts in COVID-19 convalescent patients and non-COVID-19 subjects with cardiovascular risk, compared to healthy participants. Correspondingly, we have made changes to the text to explain this, citing relevant literatures.

5) The authors could comment one or two sentences more on the interaction between T-cells and endothelial cells. What is the impact of toll-like receptors (TLRs) on the remodeling?

We have added the following sentences to Discussion section.

“It is also possible that some endothelial cells affected by SARS-CoV-2 infection had elicited pattern recognition receptors such as toll-like receptors, activating interferon pathway and inflammatory cytokine production that persist to convalescent phase. Antigen-experienced T cells preferentially adhere to activated endothelial cells and undergo transendothelial migration (Carman and Martinelli, 2015, Marelli-Berg, James et al., 2004), triggering vascular remodeling due to endothelial damage (Cuttica, Langenickel et al., 2011).”

6) If possible, this manuscript will give us more mechanistic insight if the author could add some in vitro experiments using endothelial cells and immune cell lines infected with the COVID-19 virus.

We concur with the reviewer that in vitro experimentation of endothelial-immune cell interaction under the impact of SARS-CoV-2 infection is critically needed to confirm the mechanisms. We have learnt from literature that in vitro primary cell lines or ex vivo cultivated endothelial cells are at the best moderately permissive or resistant to SARS-CoV-2 infection (PMIDs: 32386571, 33375371, 32526206). Some of the challenges that researchers have grappled with include differential susceptibility to viral infection due to endothelial heterogeneity (PMIDs: 33310781, 32091393). There have been variable evidence of endothelial dysfunction induced by mechanisms from direct infection, to indirect via exposure to viral proteins (PMID: 32587958), or through paracrine stimulation from adjacent infected cells (https://www.biorxiv.org/content/10.1101/2020.11.08.372581v1). For successful in vitro infection experimentation, it appears that 3D microenvironment in the form of organoids is required for SARS-CoV-2 infection of endothelial cells (PMID: 32333836), in line with the need to establish endothelial polarity as seen in facilitating other types of viral infections (PMID: 32847471). Due to the need to develop more advanced cellular system and BSL3 facility, we seek your kind understanding that we may not have sufficient resources to perform this experiment for the current revision. We have instead explicitly stated that our conclusions will require further supporting data.